# Immune Microenvironment Dysregulation: A Contributing Factor to Obesity-Associated Male Infertility

**DOI:** 10.3390/biomedicines13061314

**Published:** 2025-05-27

**Authors:** Rui Feng, Dexin Cheng, Wei Zhang, Jiayun Zhang, Sixiang Chen, Yan Xia

**Affiliations:** 1Department of Interventional Medicine, Zhejiang Provincial People’s Hospital (Affiliated People’s Hospital, Hangzhou Medical College), Hangzhou 310014, China; fengrui596@yeah.net (R.F.); cdx312@126.com (D.C.); 18257568898@163.com (J.Z.); 2Department of Endocrinology, Geriatric Medicine Center, Zhejiang Provincial People’s Hospital (Affiliated People’s Hospital, Hangzhou Medical College), Hangzhou 310014, China; zhangwei1@hmc.edu.cn; 3Molecular Andrology, Clinic of Urology, Pediatric Urology and Andrology, Justus-Liebig-University, 35392 Giessen, Germany

**Keywords:** obesity, male infertility, immune microenvironment, inflammatory cytokines, oxidative stress, testicular dysfunction, adipokines, blood–testis barrier, sperm quality

## Abstract

Obesity is a major contributor to male infertility, not only exacerbating infertility but also impairing the effectiveness of both surgical interventions and medical treatments. This review examines the complex relationship between obesity, the immune microenvironment, and male infertility, highlighting how obesity-induced changes in immune function lead to testicular dysfunction and impaired spermatogenesis. Key mechanisms include chronic low-grade inflammation, immune cell infiltration, and dysregulated adipokines such as leptin and adiponectin. We also explore current therapeutic strategies aimed at alleviating these effects, including lifestyle interventions, anti-inflammatory treatments, metabolic therapies, and regenerative medicine approaches, such as exosome-based therapies. Despite promising results, substantial research gaps remain, particularly in understanding the molecular mechanisms and identifying novel biomarkers for early diagnosis. Future studies should focus on multi-omics approaches, large-scale cohort studies, the gut–testis axis, and the psychological and social factors influencing male infertility. A deeper understanding of these processes is crucial for developing more effective, targeted therapies for obesity-related male infertility.

## 1. Introduction

Obesity has become a significant global health issue, and its association with male infertility is garnering increasing attention. Studies have shown that male factors account for approximately 30–50% of infertility cases, with obesity and its related metabolic abnormalities being important contributing factors [1]. The World Health Organization (WHO) estimates that about 10–15% of couples of reproductive age globally face fertility issues, with male factors contributing to approximately 50% of these cases [2,3]. Obesity is associated with a range of reproductive health issues in men, including varicocele (VC), testicular dysfunction, compromised sperm quality, and immune dysregulation, all of which significantly impact male fertility [4,5,6]. Studies indicate that 35–40% of infertile men have VC, with overweight and obese males showing a lower prevalence of left and bilateral VC, but a 63.3% increased prevalence of right VC [4,7]. Obese men have a 42% higher likelihood of having oligospermia and an 81% higher likelihood of having azoospermia compared to men with normal weight [8].

The mechanisms linking obesity to impaired male reproductive function are multifaceted, involving several biological pathways. These include hypogonadism [9], testicular heat stress and hypoxia-induced apoptosis, and endocrine disruption by obesogens [10]. The vicious cycle between obesity and hypogonadism is closely tied to metabolic changes, such as insulin resistance. Leptin’s regulatory role in this process may be compromised by resistance, which in turn affects male reproductive health and testicular function. Abdominal fat accumulation due to obesity can raise intra-abdominal pressure, affecting blood return in the pampiniform venous plexus, leading to venous congestion and dilation [11]. This can also increase scrotal temperature, inducing Deoxyribonucleic Acid (DNA) repair impairment mediated by Heat Shock Factor 1 (HSF1), and exacerbating oxidative stress and apoptosis. VC also leads to local oxidative damage in the testes [12]. The combination of these factors may jointly impair male fertility through chronic inflammation, oxidative stress, and the disruption of spermatogenesis. Moreover, endocrine disruptors in obesity-promoting diets can interfere with the hypothalamic–pituitary–gonadal axis, further impairing reproductive function [13]. 

Obesity is a metabolic disease caused by genetic and behavioral factors. In addition to being closely associated with various chronic diseases such as metabolic syndrome, hyperlipidemia, type 2 diabetes, and cardiovascular diseases, recent studies have begun to reveal the connection between obesity and immune system dysregulation [14,15,16]. The interaction between metabolism and the immune system is mediated by various factors, including immune cells, adipocytes, and the systemic inflammation induced by obesity [17]. In the context of male infertility, the immune system may mount an immune response against sperm, compromising their normal function. This immune reaction can result in decreased sperm quality, reduced motility, and even sperm death. Autoimmune factors are not uncommon in male infertility, yet the precise mechanisms underlying these immune-mediated processes remain incompletely understood [18]. The immune system’s role extends beyond mere immune responses; it interacts closely with metabolic and endocrine systems, creating a complex interplay that can profoundly affect reproductive health. Obesity-induced changes in the immune microenvironment, such as chronic inflammation and increased levels of cytokines and reactive oxygen species (ROS), can further exacerbate the detrimental effects on sperm production and function.

Although numerous systematic reviews and meta-analyses have established a significant association between obesity and male infertility—primarily highlighting the statistical correlations between increased body mass index (BMI) and impaired semen parameters (such as sperm count, motility, and morphology), as well as the dysregulation of reproductive hormones (e.g., testosterone, estradiol, inhibin B)—most studies remain confined to the epidemiological level. The underlying biological mechanisms, particularly the role of immune-related pathways, have yet to be thoroughly investigated. Based on our team’s long-term experience in clinical research on male infertility, this review focuses on the testicular immune microenvironment as a central theme. For the first time, we systematically propose a mechanistic framework wherein obesity disrupts immune homeostasis, thereby impairing spermatogenesis. We provide an in-depth analysis of how chronic low-grade inflammation—through inflammatory cytokines, adipokines, and oxidative stress—compromises immune tolerance, damages the blood–testis barrier, and inhibits sperm production. Furthermore, we integrate and summarize the regulatory roles of several key signaling pathways involved in this process, including Suppressor of Cytokine Signaling 3/Signal Transducer and Activator of Transcription 3 (SOCS3/STAT3), Focal Adhesion Kinase (FAK), Nuclear Factor Erythroid 2-Related Factor 2/Mitogen-Activated Protein Kinases (NRF2/MAPKs), and Phosphoinositide 3-Kinase/Protein Kinase B (PI3K/AKT), thereby constructing a more focused and explanatory mechanistic model distinct from the generalized listings found in previous narrative reviews. This reflects both academic originality and depth. On the basis of mechanistic insights, we also evaluate the translational potential of various intervention strategies, such as anti-inflammatory therapy, metabolic improvement, and advanced regenerative approaches involving exosomes. Finally, we propose that future studies should emphasize multi-omics integration, large-scale prospective cohort studies, and emerging directions such as the “gut–testis axis”, to support the development and optimization of precision medicine approaches for male infertility.

The structure of this review is as follows: Section 2 introduces the physiological role of the testicular immune microenvironment; Section 3 discusses the specific immune dysregulations induced by obesity; Section 4 integrates basic and clinical evidence to elucidate the causal links among obesity, immune imbalance, and male infertility; and the final section presents therapeutic strategies targeting immune mechanisms and outlines future research directions aimed at improving male reproductive health under the precision medicine paradigm.

## 2. Immune Microenvironment and Male Reproductive Health

### 2.1. Normal Immune Microenvironment in the Male Reproductive System

#### 2.1.1. Overview of Immune Cells and Their Functions in the Testes 

The testis, as an important organ of the male reproductive system, primarily functions in sperm production and the synthesis of male hormones [19]. Figure 1 illustrates the structure of the testis, including its immune microenvironment. The testis hosts a unique immune microenvironment composed of various immune cells, such as macrophages, dendritic cells (DCs), and T lymphocytes, situated in the interstitial space [20]. These cells play a critical role in maintaining immune tolerance and supporting normal testicular function under both physiological and pathological conditions by sustaining an immunosuppressive environment that prevents autoimmune attacks on germ cells [21].

The testis is composed of seminiferous tubules, interstitial cells, and a complex network of blood vessels and immune cells that support sperm production and hormone synthesis. Immune regulation in the testis is maintained through the production of immunosuppressive factors, which support immune privilege and promote germ cell survival. In rodent testes, resident macrophages, tolerogenic dendritic cells, T cells, and mast cells in the interstitium interact with interstitial cells, Sertoli cells, and peritubular cells to release factors that create a favorable microenvironment. Abbreviations: PTCs (peritubular myoid cells); LCs (Leydig cells); TCs (telocytes); TFs (Testicular fibroblasts); iTMs (Interstitial macrophages); PTMs (Peritubular macrophages); DCs (dendritic cells); SPG (spermatogonia); SPC (spermatocyte); TJ (tight junction); SPT (spermatid); SPZ (spermatozoa); SCs (Sertoli cells); T cell (T lymphocyte).

Macrophages are immune cells with phagocytic characteristics, capable of ingesting foreign substances, pathogens such as bacteria, and clearing cell debris. In the testis, they account for about 20% of the interstitial cells, making them one of the most abundant and heterogeneous immune populations [22]. Under normal conditions, most testicular macrophages (TMs) express core markers such as F4/80, cluster of differentiation (CD)-11b, apoptosis-inducing factor (AIF), and C-X3-C Motif Chemokine Receptor 1 (CX3CR1) [23]. Their heterogeneity is evident from the differential expression of surface markers like CD64 and major histocompatibility complex class II (MHC II) [24]. Two main types of TM exist: interstitial (iTMs) and peritubular (pTMs) macrophages. The former are located in the testicular interstitium, while the latter are near the basal membrane of the seminiferous tubules. iTMs, which are M2-type, express immunosuppressive cytokine Interleukin (IL)-10, whereas pTMs, an M1-type, produce high levels of pro-inflammatory cytokines such as IL-1β [25]. The developmental origins of these TMs are not fully understood: iTMs may derive from embryonic progenitors, while pTMs are thought to originate from bone marrow-derived cells [23]. The testicular microenvironment contains various immune-regulatory molecules, including testosterone, prostaglandins, corticosterone, activin, and 25-hydroxycholesterol (25-HC), which likely influence macrophage phenotype and function [22]. TMs also display immunosuppressive functions, secreting high levels of IL-10 and producing low levels of tumor necrosis factor-alpha (TNF-α) and nitric oxide (NO) upon lipopolysaccharide (LPS) stimulation. However, excessive TNF-α and NO production can compromise the immune privilege of the testis, impairing spermatogenesis and steroidogenesis [26,27]. Additionally, TMs suppress T cell activation and proliferation, promoting the differentiation of naïve T cells into regulatory T cells (Tregs) [28]. Following testicular inflammation (such as after acute LPS stimulation or experimental chronic autoimmune epididymo-orchitis), TM numbers increase, playing a key role in local immune responses and tissue repair [29,30,31].

DCs are professional antigen-presenting cells (APCs) involved in both innate and adaptive immune responses, playing a role in T cell activation and tolerance induction. In the interstitial cells of normal testes, DCs account for about one-tenth of the macrophage population [27]. CD103^+^ DCs expressing MHCII and co-stimulatory molecules such as CD80 and CD86 have been observed in both normal rat testes and testicular draining lymph nodes (TLNs) [32]. Compared to inflamed testes, DCs in normal testes show lower levels of C-C chemokine receptor (CCR)7 and IL12 subunit p35 messenger RNA (Il12p35 mRNA), indicating an immature state [33]. CCR7 is involved in the migration of DCs to the LNs, while Il12p35 promotes T helper 1 cell (Th1) immune responses. The low expression levels of these molecules in immature DCs suggest they lack the ability to trigger a strong immune response, thereby contributing to the maintenance of testicular immune tolerance and immune privilege. When cultured with Sertoli cells, DCs show reduced expression of surface molecules such as MHCII, CD80, CD86, CCR7, and CD11c, indicating lower immunogenicity [34]. Furthermore, these DCs exhibit enhanced immunoregulatory capabilities, such as inhibiting T cell proliferation and promoting the generation of Foxp3^+^ Tregs, thereby preventing the immune system from attacking normal testicular tissue [34].

T cells differentiate into effector or regulatory lineages depending on environmental signals, exerting pro-inflammatory or immune-tolerant effects, ultimately leading to either inflammation or immune tolerance [35]. In the testes, CD4^+^, CD8^+^ T cells, CD8^+^ memory T cells, and Foxp3^+^ Tregs have been identified. Both CD4^+^ and CD8^+^ T cells release pro-inflammatory cytokines, such as TNF-α, Interferon gamma (IFN-γ), and Fas Ligand (FasL), with CD8^+^ T cells being the primary producers of FasL and TNF-α. Under normal conditions, occasional IL-4-expressing T cells are found in the testes [19]. Foxp3^+^ Tregs are present in normal testes, most of which exhibit a memory phenotype and produce transforming growth factor beta (TGF-β). Tregs are activated by antigens from the seminiferous tubules, triggering proliferative responses and suppressing conventional T cell proliferation, thereby maintaining a tolerogenic environment [36]. Under inflammatory conditions, pathogenic T cells may overwhelm the suppressive function of Tregs, leading to autoimmune responses. Despite the increased suppressive capacity of Tregs from the TLN, they are still unable to prevent germ cell attack [37].

Mast cells (MCs) are primarily localized around the seminiferous tubules in the testis, with an increased number observed under inflammatory conditions, such as spermatogenesis defects, VC, infertility, experimental autoimmune orchitis (EAO), and testicular torsion. The tryptase released by MCs promotes fibroblast proliferation and collagen synthesis, leading to fibrosis of the seminiferous tubules [38].

#### 2.1.2. Blood–Testis Barrier (BTB) and Immune Privilege in Testes

The immune privilege of the testes is maintained by its unique physical structure, systemic immune tolerance, and local immune-suppressive environment, which together prevent immune reactions against germ cells. The BTB, formed by tight junctions between Sertoli cells, divides the seminiferous epithelium into basal and adluminal compartments, and by isolating late-stage germ cells from immune cells, it maintains immune privilege (Figure 2) [39]. The BTB also supports the transition of spermatogonia to sperm and facilitates the movement of pre-leptotene spermatocytes from the basal to the adluminal compartment, a key step in spermatogenesis [40]. Bioactive peptides and molecules secreted by testicular cells regulate the BTB function, further supporting spermatogenesis.

Inside the testis, the seminiferous tubules contain spermatogonia and other stages of meiotic cells. Sertoli cells lining these tubules create the BTB by establishing tight junctions between their cytoplasmic extensions, which regulate permeability and help maintain immune privilege. The abbreviations used are as follows: BTB (blood–testis barrier); mTORC1 (Mechanistic Target of Rapamycin Complex 1); mTORC2 (Mechanistic Target of Rapamycin Complex 2); and FAK (Focal Adhesion Kinase).

The BTB limits the entry of immune cells and antibodies into the seminiferous tubules and, through lactate dehydrogenase isoenzyme (LDHc4), reversely crosses the barrier, forming immune complexes to promote immune tolerance [41]. The dynamic regulation of the BTB is controlled by mechanistic Target of Rapamycin (mTOR) and FAK. Mechanistic Target of Rapamycin Complex (mTORC)1 increases its permeability, while mTORC2 strengthens it. Phosphorylation sites of FAK regulate the “leakiness” and stability of the BTB [42,43]. However, environmental pollutants such as molybdenum, cadmium, nickel, arsenic, etc., can damage the BTB through oxidative stress, leading to testicular toxicity and reproductive dysfunction. The BTB also allows for xenografts or allografts in the testicular interstitial space to survive long term, indicating that the testes’ immune privilege is not solely dependent on the structural barrier provided by the BTB [44,45].

### 2.2. Immune Dysregulation and Male Infertility

#### 2.2.1. Autoimmune Infertility 

When the immune system mistakenly identifies sperm as foreign invaders, it produces anti-sperm antibodies (ASAs), triggering an immune response [46]. Under normal conditions, sperm are protected from immune surveillance by the BTB [47]. However, when the BTB is compromised due to injury, surgery, infection, or congenital defects, sperm antigens become exposed, leading to ASA production [48,49,50]. ASAs are an important cause of male immune infertility, with studies showing a prevalence of 3.4% in a large-scale retrospective study [51]. 

ASAs negatively affect semen parameters, including reduced sperm motility and sperm aggregation. They bind to apoptosis-related proteins such as caspase-3 and hsp70, promoting sperm apoptosis [52]. Additionally, ASAs hinder sperm passage, inhibit fertilization, and affect early embryonic development [53,54,55]. While both IgA- and IgG-type ASAs are commonly present, IgA is more clinically significant, whereas IgG is more effective in activating the complement system, leading to sperm lysis [56]. While not all ASAs affect fertility significantly, their impact varies by semen heterogeneity and gender differences [57]. In addition, when the immune system mistakenly recognizes testicular or epididymal tissue as foreign, especially in cases of genital tract infections, surgical injuries, or trauma, it can lead to chronic inflammation and dysfunction of the testicular or epididymal tissues, causing spermatogenesis disorders, hormonal imbalances, and infertility [58]. 

#### 2.2.2. Failure of Immune Tolerance

The male reproductive system protects sperm from immune attacks through immune tolerance mechanisms. Under conditions such as trauma, infection, chronic inflammation, autoimmune diseases, and environmental factors, this mechanism may fail, leading to immune cells attacking sperm or reproductive tissues, which can result in infertility. In Treg-depleted mice, mononuclear phagocytes (MPs) project towards and invade the epididymal lumen, leading to a decrease in sperm count and motility, and causing severe fertility defects [59].

#### 2.2.3. Chronic Inflammation

Local inflammation, such as chronic inflammation in the male reproductive system or reproductive tract infections, can lead to an excessive release of inflammatory cytokines (e.g., TNF-α, IL-6, IL-1β), disrupting spermatogenesis and inducing oxidative stress. This, in turn, causes sperm DNA fragmentation (SDF), reduced motility, and even programmed cell death (apoptosis) [60]. Environmental factors, lifestyle habits, and other factors may contribute to the abnormal activation of the immune system, exacerbating this process. Studies have shown that up to 15% of male infertility cases are linked to infections and inflammatory conditions [61]. Additionally, systemic inflammatory response syndrome (SIRS) may exacerbate fertility decline by affecting sperm function. Factors such as obesity, diabetes, and systemic infections can induce a systemic inflammatory state, negatively impacting the reproductive system [62,63]. Studies have shown that increased ROS and oxidative stress are the main causes of male infertility induced by diabetes [64]. Low levels of ROS play a positive regulatory role in sperm function, including promoting spermatogenesis, capacitation, maturation, and fertilization [65]. However, excessive ROS can increase the antioxidant capacity of sperm, leading to oxidative damage. High levels of ROS attack polyunsaturated fatty acids in the sperm cell membrane, causing lipid peroxidation, which disrupts the lipid bilayer of the cell membrane and affects the fluidity of the sperm membrane, material transport, energy metabolism, and immune defense [66]. Oxidative stress damages sperm nuclear and mitochondrial DNA, accelerates sperm apoptosis, and reduces sperm count and semen quality [67]. Clinical evidence shows that the levels of DNA oxidation and apoptosis in sperm from infertile men are significantly higher than in normal men [68]. Hyperglycemia leads to elevated ROS levels, inhibiting the PI3K/AKT and mTOR signaling pathways, which in turn induces abnormal autophagy in testicular cells [64]. The degradation of the autophagy-related protein P62 inhibits the activation of NRF2, weakening the antioxidant capacity of the testis, exacerbating oxidative stress, and creating a vicious cycle [69]. Furthermore, the effects of autophagy on male reproductive function are manifested in several aspects: Leydig cell dysfunction leading to decreased testosterone secretion, inhibited Sertoli cell proliferation affecting spermatogenesis, BTB disruption weakening immune protection, PI3K/AKT/mTOR signaling inhibition resulting in sperm deformities and decreased motility, and impaired spermatogonia proliferation affecting spermatogenesis [69,70,71,72,73].

## 3. Obesity and Its Impact on the Immune System

### 3.1. Chronic Low-Grade Inflammation 

Chronic low-grade inflammation induced by obesity is a key driver of various metabolic disorders, particularly playing a significant role in insulin resistance and vascular dysfunction. The following discusses how obesity activates inflammatory responses through multiple signaling pathways, leading to the onset of chronic low-grade inflammation.

In obesity, the enlargement of adipose tissue, particularly abdominal fat, is a major contributor to chronic low-grade inflammation. As adipocytes increase in size, cellular metabolic activity intensifies, leading to higher levels of stress, including oxidative stress and endoplasmic reticulum (ER) stress. These stressors activate key inflammatory pathways such as Nuclear Factor kappa-light-chain-enhancer of activated B cells (NF-κB) in the liver, promoting the production of pro-inflammatory cytokines like IL-6, IL-1β, and TNF-α, which together foster insulin resistance and low-grade inflammation [74]. Furthermore, both metabolic and ER stress can generate ROS, which further enhance the expression of TNF-α in Tamm-Horsfall Protein-1 (THP-1) cells through ROS/ C/EBP homologous protein (CHOP)/Hypoxia-Inducible Factor 1 alpha (HIF-1α) and MAPK/NF-κB-dependent mechanisms [75]. High-fat diets (HFDs) exacerbate this process by activating the Jun N-terminal kinases (JNK), TNF-α, and NF-κB signaling pathways, thereby amplifying inflammation and insulin resistance. Interestingly, raspberry treatment has been shown to reduce the levels of TNF-α, IL-1β, IL-6, and the phosphorylation of JNK in HFD mice, improving insulin sensitivity in skeletal muscle via AMP-activated protein kinase-α1 activation [76]. Additionally, SOCS proteins, which negatively regulate cytokine signaling in the Janus Kinase (JAK)/STAT pathway, play a significant role in promoting inflammation in obesity. The upregulation of SOCS3 and SOCS1 in tissues like muscle, liver, and adipose further inhibits insulin signaling while exacerbating inflammation by stimulating p38MAPK and STAT3 phosphorylation, without affecting anti-inflammatory cytokines [77]. In Toll-Like Receptor 4 (TLR4) mice, HFD-induced obesity activates phosphorylated Akt (pAkt) in the liver, with TLR4 signaling—via NF-κB activation—leading to palmitic acid-induced vascular inflammation and insulin resistance, disrupting endothelial NO signaling [78]. Lastly, noncanonical Wingless/Int (Wnt) signaling, particularly through Wnt5a, has been linked to increased adipose tissue inflammation and insulin resistance, promoting the release of pro-inflammatory cytokines independent of adipose tissue expansion [79]. In summary, obesity-induced inflammation is a multifaceted process involving the activation of multiple signaling pathways, including NF-κB, JAK/STAT, MAPK, and noncanonical Wnt signaling. 

### 3.2. Adipokines 

Adipokines are peptides that convey the functional status of adipose tissue to organs like the brain, liver, pancreas, immune system, blood vessels, and muscles, regulating energy balance, fat metabolism, inflammation, and other physiological processes through interactions with various cells. Their secretion is altered when adipose tissue becomes dysfunctional [80]. 

Under physiological conditions, circulating leptin levels are positively correlated with white adipose tissue (WAT) mass. In overweight and obese individuals, leptin levels increase significantly due to the expansion of adipose tissue, and it is recognized as a crucial pro-inflammatory adipokine [81]. In the central nervous system, leptin regulates energy homeostasis by inducing anorexigenic factors and inhibiting orexigenic factors, thereby reducing food intake and promoting energy expenditure [82]. Given its dual nature as both a hormone and cytokine, leptin serves as a critical link between the neuroendocrine and immune systems [83]. In the immune system, leptin modulates both innate and adaptive immune responses through multiple mechanisms. It promotes monocyte and macrophage activation, enhances neutrophil chemotaxis, and stimulates natural killer (NK) cell development and pro-inflammatory cytokine release. In adaptive immunity, leptin promotes the proliferation of naïve T and B cells, suppresses Treg cell differentiation, and drives the polarization of TH1 cells toward a pro-inflammatory phenotype while increasing T helper 17 cell (TH17) proliferation and reactivity [84]. Leptin plays a time- and dose-dependent role in regulating immune cell function and metabolism, with its specific mechanisms remaining unclear under different experimental conditions. Resistin and apelin are thought to promote inflammation by inducing the secretion of pro-inflammatory cytokines [85,86]. 

On the contrary, adiponectin has anti-inflammatory and immune-regulatory effects, but its plasma levels are significantly reduced in obesity, weakening its anti-inflammatory effects and leading to worsened chronic low-grade inflammation and metabolic disorders [87]. Adiponectin also affects the function of myeloid monocytes, which are key regulators of innate immunity, and negatively impacts the phagocytic activity of macrophages [88]. The dysregulation of adipokines, particularly the imbalance between pro-inflammatory and anti-inflammatory adipokines, is a central mechanism underlying obesity.

### 3.3. Immune Cell Infiltration 

Under normal conditions, the immune environment in adipose tissue is maintained by anti-inflammatory cells such as Tregs and M2 macrophages to ensure tissue homeostasis. However, in obesity, the infiltration of immune cells significantly increases, and at the same time, the metabolism of these cells undergoes a shift, driving immune cells to transition from an oxidative phosphorylation-dependent metabolic mode to one that prefers glycolysis. For example, macrophages shift from the anti-inflammatory M2 phenotype to the pro-inflammatory M1 phenotype, secreting more pro-inflammatory cytokines (such as TNF-α and IL-6). This process is partially mediated through the monocyte chemoattractant protein-1 (MCP-1)/ CCR2 pathway derived from hypertrophic adipocytes, which promotes the accumulation of more macrophages in the adipose tissue of obese individuals [89]. Additionally, the increased extracellular lipid concentration in the local environment is a key mechanism leading to macrophage accumulation, inducing lipid metabolism disorders in macrophages and exacerbating inflammatory responses. In the early stages of obesity, neutrophils rapidly infiltrate adipose tissue, induced by IL-8 secreted by adipocytes and macrophages, as well as lipid peroxidation products. They produce ROS and neutrophil elastase (NE), amplifying oxidative stress and sustaining the pro-inflammatory environment [90,91]. The long-term upregulation of high levels of leptin and MHC II molecules promotes the differentiation of CD4^+^ T cells and CD8^+^ T cells into pro-inflammatory subtypes (such as Th1 and Th17) [92], which alters the function of APCs and adipose tissue macrophages, enhancing the inflammatory response and inhibiting the function of Tregs, thus leading to immune dysregulation and the suppression of immune defense against infections [93]. Obesity also promotes the infiltration of B cells into visceral adipose tissue and activates T cells through MHC II-mediated antigen presentation, which further enhances the activation of M1 macrophages and the inflammatory response in adipose tissue [94]. Moreover, obesity promotes the formation of a specific NK cell subpopulation through the IL-6/Stat3 signaling pathway, which expresses IL6Ra and Colony Stimulating Factor 1 Receptor (Csf1r), contributing to the onset of obesity, insulin resistance, and associated inflammation [95]. Overall, obesity alters the immune environment through multiple mechanisms, driving the infiltration, metabolic changes, and functional alterations of immune cells. These changes together promote chronic inflammatory responses in adipose tissue.

## 4. Intersection of Obesity, Immune Microenvironment, and Male Infertility

Obesity can directly or indirectly lead to male reproductive dysfunction by altering the immune microenvironment of the testes. The immune microenvironment of the testes plays a key role in maintaining immune tolerance and tissue homeostasis. However, chronic low-grade inflammation, immune cell imbalance, and metabolic abnormalities induced by obesity disrupt this balance, increasing the risk of male infertility (Table 1). 

This table summarizes the impact of obesity on testicular function through the modulation of inflammatory cytokines, oxidative stress, hormonal imbalance, and adipokine dysregulation. These mechanisms disrupt the testicular immune microenvironment, impair spermatogenesis, and lead to reduced sperm quality and hormonal dysfunction. Abbreviations: IL-6 (Interleukin-6); TNF-α (Tumor Necrosis Factor-alpha); SOCS3 (Suppressor of Cytokine Signaling 3); STAT3 (Signal Transducer and Activator of Transcription 3); NF-κB (Nuclear Factor kappa-light-chain-enhancer of activated B cells); NLRP3 (NOD-like receptor family, pyrin domain containing 3); BTB (blood–testis barrier); ZO-1 (Zonula Occludens-1); CX43 (Connexin 43); PI3K (Phosphoinositide 3-Kinase); cAMP-PKA (Cyclic Adenosine Monophosphate-Protein Kinase A); RBP4 (Retinol-Binding Protein 4); APCs (Antigen-Presenting Cells); CD4^+^ (Cluster of Differentiation 4 Positive); VC (Varicocele); NR3C1 (Nuclear Receptor Subfamily 3 Group C Member 1); MTHFR (Methylenetetrahydrofolate Reductase); IL-1 (Interleukin-1); Zn (Zinc); Se (Selenium); Cu (Copper); Fe (Iron); SDF (Sperm DNA Fragmentation); DHT (Dihydrotestosterone); ROS (reactive oxygen species).

### 4.1. Inflammatory Cytokines in Obesity and Their Role in Male Infertility

With the increase in body fat tissue, chronic low-grade inflammation induced by obesity becomes a major threat to male reproductive health. This alteration of the immune microenvironment, particularly the excessive secretion of pro-inflammatory cytokines, may disrupt the immune tolerance mechanisms of the testes, thereby impairing the function of the reproductive system and leading to male infertility.

Obesity increases the secretion of IL-6 and other pro-inflammatory cytokines from macrophages, Leydig cells, and Sertoli cells in the testis, further exacerbating testicular inflammation [96]. Additionally, the elevated IL-6 induced by obesity activates the SOCS3/STAT3 signaling pathway, inhibiting the expression of Zinc Finger Protein 637 (Zfp637) and subsequently downregulating SRY-box transcription factor 2 (SOX2) expression, thereby disrupting spermatogonial cell differentiation in mice [97]. Obesity increases TNF-α levels in multiple tissues, including the testis, where it plays a crucial role in inflammation [14]. Elevated TNF-α impairs spermatogenesis by inducing germ cell apoptosis, disrupting Sertoli cell junctions, and inhibiting steroidogenesis in Leydig cells [98]. It also disrupts Ca^2+^ homeostasis, which is essential for sperm function, leading to reduced motility and abnormal morphology [99]. A study of seminal plasma revealed a negative correlation between TNF-α and sperm motility and morphology [100]. TNF-α activates NF-κB, a transcription factor involved in immune response regulation and germ cell apoptosis, contributing to male infertility [101]. Elevated TNF-α and NF-κB expression in obese mice testes correlated with impaired sperm function and lower pregnancy rates in mated females. Semen samples from 272 male donors confirmed a correlation between obesity, sperm quality, and pro-inflammatory cytokine levels, showing elevated IL-6 and TNF-α in the semen of obese men, along with reduced sperm concentration and motility [102].

Chronic inflammation, characterized by elevated TNF-α, IL-1β, and IL-6 levels, activates the IκB Kinase Beta (IKKβ)/NF-κB pathway. This leads to testicular structural damage, reduced testosterone levels, and decreased sperm motility [103]. Studies have shown that a high-fat diet significantly increases the levels of the NLRP3 inflammasome and pro-inflammatory cytokines (such as IL-6 and TNF-α) in the testis, epididymis, prostate, and seminal vesicles, with a corresponding increase in these cytokines in the serum [102]. In rats on a high-fat diet, the elevation of inflammation markers like High Mobility Group Box 1 (HMGB1) and NLRP3 in the testes is closely associated with a reduction in sperm count, a decrease in seminiferous tubule diameter and epithelial height, and a decline in Johnsen score [104]. Additionally, elevated corticosterone and IgG levels in the serum of obese mice indicated that the inflammatory response induced by obesity is closely related to immune regulation. Corticosterone has potent immunosuppressive and anti-inflammatory effects, while IgG plays a role in the inflammatory immune response [105]. Furthermore, obesity exacerbates the pathological effects of VC, a condition that significantly increases levels of ASAs and pro-inflammatory cytokines such as IL-1 and TNF-α. These cytokines activate the NLRP3 inflammasome pathway, disrupting immune tolerance in the testis and aggravating male infertility [4].

The gut microbiota critically regulates host metabolism, immunity, and endocrine function, directly impacting male reproductive health. It synthesizes essential micronutrients (vitamin A, K, folate, calcium) vital for testicular function, spermatogenesis, and BTB integrity [106,107]. HFD-induced gut dysbiosis in obesity disrupts nutrient synthesis and promotes inflammation. Fecal microbiota transplantation (FMT) from HFDs fed to normal mice reproduces gut imbalance, impairing Leydig cells, BTB integrity, and spermatogenesis [108]. Intervention with catalpol, an anti-inflammatory and antioxidant agent, partially restores gut microbiota diversity and improves testicular function, suggesting that gut dysbiosis-mediated oxidative stress and inflammation contribute to reproductive dysfunction [109]. Dysbiosis has been associated with decreased serum testosterone levels, and studies in germ-free mice have shown a downregulation of genes related to steroidogenesis and BTB formation—such as occludin, Zonula Occludens-2 (ZO-2), and E-cadherin—highlighting the essential role of balanced gut microbiota in maintaining endocrine and barrier functions in the testes [110].

In obese men, variations in the NR3C1 gene are associated with a reduction in sperm count. The NR3C1 gene, which is responsible for regulating the body’s response to cortisol, can indirectly affect sperm quality by modulating stress and inflammatory responses [111]. When the MTHFR gene is mutated, it disrupts the one-carbon metabolism process, leading to a decrease in the number of methyl donors. This change alters the DNA methylation status of genes closely related to inflammation, such as TNF-α and IL-6, thereby activating an excessive immune response [112]. Moreover, the mutation of the MTHFR gene increases the level of homocysteine, triggering an oxidative stress response and promoting the release of inflammatory factors, which further exacerbates the imbalance of the immune microenvironment [113]. Under the combined action of factors like obesity, chronic low-grade inflammation is further aggravated, which inhibits sperm production and reduces sperm quality, ultimately leading to male infertility.

### 4.2. Oxidative Stress, Hormonal Imbalance, and Testicular Dysfunction Induced by Obesity

Chronic low-grade inflammation induced by obesity leads to the excessive generation of ROS, which damage Leydig cells, impair testosterone bioavailability, and promote germ cell apoptosis [114]. Additionally, oxidative stress disrupts the BTB by reducing the expression of key proteins such as zonula occludens-1 (ZO-1), osteocalcin (OCN), CX43, N-Cadherin (N-CAD), and Vang-like 2 (VANGL2), impairing the transport of spermatogonia and spermatogenesis [115]. Moreover, excessive ROS damage sperm membranes and DNA, triggering lipid peroxidation and exacerbating SDF [116]. In the obese state, the antioxidant system in the testes becomes weakened [117], unable to effectively neutralize excess ROS. This overwhelms the antioxidant defense system in the genital tract, leading to oxidative stress, which negatively affects sperm concentration, motility, and progressive motility [118]. 

Furthermore, obesity triggers inflammation in adipose tissue, which increases the expression of aromatase in adipocytes through Peroxisome Proliferator-Activated Receptor Gamma (PPARγ) and pro-inflammatory factors. This leads to the excessive conversion of androgens into estrogens, disrupting the hormonal balance in the male body and impairing the spermatogenic function of the testis [119]. Finally, FAK interferes with the dynamic regulation of the BTB through Proto-oncogene tyrosine-protein kinase Src (Src) kinase phosphorylation, while the dysregulation of the NRF2/MAPKs signaling pathway exacerbates BTB disruption and germ cell apoptosis, further impairing the immune-privileged microenvironment and ultimately affecting normal sperm production [120]. 

Trace elements such as zinc (Zn), copper (Cu), iron (Fe), and selenium (Se) serve as essential cofactors for key antioxidant enzymes, including superoxide dismutase (SOD) and glutathione peroxidase (GPx). These enzymes play a pivotal role in maintaining redox homeostasis and scavenging ROS, thereby protecting spermatozoa from oxidative damage. However, under obese conditions, deficiencies in trace elements—particularly Zn and Se—can impair antioxidant enzyme activity, weaken the oxidative defense system, and contribute to a vicious cycle of oxidative stress and inflammation [121,122,123]. Obesity-induced dysfunction or altered expression of zinc transporters (ZIPs), such as ZIP family members (e.g., ZIP12), may disturb both systemic and testicular zinc levels, compromising sperm quality and fertilization capacity [124]. Zinc, as a critical cofactor of zinc-α2-glycoprotein (ZAG), is indispensable for ZAG’s anti-inflammatory and lipolytic functions [125]. Deficiency of Zn impairs these activities, contributing to metabolic dysregulation. Additionally, trace elements such as Zn, nickel (Ni), and cobalt (Co) can modulate the expression of immune-related cytokines in adipocytes, thereby attenuating obesity-induced inflammation [126]. Moreover, the impact of trace elements on reproductive function exhibits a clear dose-dependent effect. For instance, molybdenum (Mo) at an appropriate concentration (e.g., 25 mg/L) may improve sperm quality, whereas excessive exposure (≥100 mg/L) exacerbates oxidative damage [127]. 

### 4.3. The Role of Adipokines in Testicular Function

Adipokines are crucial mediators between obesity and the testicular immune microenvironment, and their abnormal secretion significantly impacts testicular function. Obesity-induced hyperleptinemia binds to receptors on testicular seminiferous tubule cells, activates the PI3K pathway, increases free radical production, and inhibits antioxidant enzyme activity, leading to oxidative stress. This damages seminiferous tubule cells and sperm DNA, resulting in cell apoptosis, DNA fragmentation, reduced sperm count, abnormal morphology, and seminiferous tubule atrophy [128]. Moreover, hyperleptinemia upregulates SOCS3, inhibiting JAK2 phosphorylation and blocking leptin signaling, impairing testicular function [129]. Leptin regulates testicular function via the JAK2/STAT3 pathway, promoting the transcription of steroidogenic genes in Leydig cells. However, the upregulation of SOCS3 inhibits this process, leading to decreased testosterone levels [130].

Adiponectin and its receptors (AdipoR1 and AdipoR2), which are expressed in Leydig cells and seminiferous tubules, suggest its involvement in testicular function [131]. Adiponectin promotes testosterone synthesis through the cAMP-PKA pathway and has anti-inflammatory and metabolic regulatory effects. In obesity, decreased adiponectin levels reduce its protective role, increasing the risk of inflammation and fibrosis in the testes. Restoring adiponectin levels can improve testicular inflammation, oxidative stress, and steroidogenesis. Similar results were observed in diabetic mice, where adiponectin replacement therapy improved testicular function, including steroidogenesis, energy substrate transport, and reduced oxidative stress markers [132].

Retinol-binding protein 4 (RBP4) is a multifunctional protein that serves both as a transporter of retinol and as an adipokine. Under obese conditions, elevated RBP4 secretion from adipose tissue activates APCs and macrophages, promoting the release of pro-inflammatory cytokines and driving CD^4+^ T cell activation. This chronic low-grade inflammation not only exacerbates systemic insulin resistance, but may also impair testicular immune homeostasis by disrupting the BTB or interfering with Sertoli cell function [133]. Experimental studies further demonstrate that excessive RBP4 directly suppresses the transcription and translation of key steroidogenic enzymes (3β-hydroxysteroid dehydrogenase (3β-HSD) and steroid 5 alpha-reductase type 1 (SRD5A1)) in Sertoli cells, leading to reduced testosterone and dihydrotestosterone (DHT) synthesis, along with downregulated androgen receptor (AR) expression. Conversely, RBP4 knockdown produces the opposite effects, significantly enhancing steroidogenic enzyme activity and androgen secretion [134]. Thus, obesity-associated RBP4 elevation contributes to male reproductive dysfunction through dual mechanisms: indirectly via systemic inflammation and directly by inhibiting steroidogenesis and AR signaling pathways. These findings establish RBP4 as a critical pathological mediator linking metabolic disorders to male infertility. Figure 3 illustrates the relationship between obesity, immune microenvironment dysregulation, and male infertility.

## 5. Obesity and Male Infertility: Immune Microenvironment Intervention and Therapeutic Strategies

### 5.1. Lifestyle Interventions 

Lifestyle modification constitutes the cornerstone of managing obesity-related male infertility. Weight reduction through structured exercise and dietary adjustment has been consistently associated with significant improvements in sperm parameters, including concentration, motility, and morphology in obese men [135,136]. Nutritional strategies emphasizing polyunsaturated fatty acids, dietary fiber, and antioxidants exert anti-inflammatory effects and contribute to the restoration of endocrine homeostasis, thereby enhancing reproductive function [137,138]. These non-pharmacologic approaches provide a foundation upon which more targeted therapeutic interventions can be developed.

### 5.2. Integrated Anti-Inflammatory and Metabolic Therapies: From Pharmacological Agents to Microbiota-Derived Metabolites

Pharmacological strategies complement lifestyle modifications by targeting the chronic inflammation and metabolic dysregulation inherent in obesity. Anti-inflammatory agents, including TNF-α inhibitors and IL-6 antagonists—originally developed for autoimmune and inflammatory disorders—have demonstrated efficacy in preclinical models in attenuating testicular inflammation and improving spermatogenic function [139,140]. 

Metabolic regulators such as metformin and glucagon-like peptide-1 (GLP-1) receptor agonists offer additional benefits. Metformin enhances insulin sensitivity, reduces testicular inflammation, increases testosterone production, and improves sperm motility [141,142]. GLP-1 receptor agonists, such as semaglutide, have been shown to restore redox homeostasis and mitigate oxidative injury in testicular tissues, thereby contributing to the recovery of reproductive function [143,144]. Furthermore, therapeutic strategies aimed at increasing adiponectin levels may offer dual anti-inflammatory and metabolic benefits, given its crucial role in energy homeostasis and reproductive physiology [145,146]. 

Elevated oxidative stress in obese individuals is associated with decreased activity of antioxidant enzymes, impaired mitochondrial membrane potential in spermatozoa, and compromised sperm quality [147]. While supplementation with antioxidants such as coenzyme Q10 and omega-3 fatty acids has been reported to reduce DNA fragmentation index (DFI) [148,149], its effects on seminal oxidative-reduction potential (sORP) remain limited [150]. These findings suggest that antioxidant therapy alone may be insufficient to fully reverse redox imbalance in the male reproductive tract. 

Recent advances highlight the pivotal role of the gut–testis axis in regulating male fertility under metabolic stress. Short-chain fatty acids (SCFAs), particularly acetate and butyrate—produced by gut microbiota—exert anti-inflammatory and antioxidative effects. Acetate has been shown to activate the NRF2/peroxisome proliferator-activated receptor gamma (PPAR-γ) signaling cascade, thereby attenuating oxidative stress and inflammation in obese rodents [151]. Butyrate restores key metabolic pathways associated with steroidogenesis, contributing to improved sperm quality in animal models [152]. Future drug development may focus on several key areas, including SCFA-targeted delivery systems, NRF2/PPAR-γ pathway-specific agonists, and metabolic reprogramming regulators, employing multi-target intervention strategies to address obesity-related male infertility. However, clinical translation still requires overcoming critical challenges such as optimizing drug delivery efficiency and personalizing treatment regimens.

Natural bioactive compounds such as resveratrol (RSV) and icariin (ICA) have also demonstrated therapeutic potential. Both compounds modulate the gut microbiota, suppress the activation of the NLRP3 inflammasome, and restore testicular steroidogenesis, providing multi-targeted benefits for reproductive and metabolic health [153,154]. Their synergistic effects highlight the promise of microbiota-targeted nutraceuticals in the management of obesity-induced male infertility. 

Additionally, bacterial contamination of semen—commonly involving *Escherichia coli*, *Staphylococcus haemolyticus*, and *Bacteroides ureolyticus*—can induce leukocyte activation and excessive production of reactive oxygen intermediates (ROIs). This promotes lipid peroxidation of sperm membranes and contributes to reduced sperm motility and function [155]. Therapeutic strategies aimed at limiting leukocyte-driven ROS generation may serve as adjunctive measures in reducing oxidative sperm damage. 

### 5.3. Regenerative Medicine and Biologic Therapies

In the field of regenerative medicine and biologic therapies, exosomes derived from mesenchymal stem cells have drawn much attention for their immunomodulatory and anti-inflammatory properties, among other properties, and lack of risk of stem cell transplantation. Exosomes and mesenchymal stem cells are potential therapies for improving testicular function. They can regulate the local immune microenvironment of the testes, repair testicular inflammation and damage caused by obesity, promote spermatogenesis, restore normal testicular function, and offer new treatment ideas for male infertility related to obesity [156].

## 6. Future Directions and Research Gaps

Although significant progress has been made in elucidating the relationship between obesity, the immune microenvironment, and male infertility, many critical questions remain unanswered. These knowledge gaps hinder the development of effective diagnostic tools and therapeutic interventions. Future research should prioritize the following directions, all centered on deepening our understanding of immune dysregulation in obesity-related male infertility.

### 6.1. Mechanistic Insights and Biomarker Discovery

Currently, the mechanisms by which obesity alters the testicular immune microenvironment remain incompletely understood. Future research leveraging multi-omics approaches, including transcriptomics, metabolomics, and proteomics, are promising tools to reveal how obesity-induced metabolic stress reprograms immune signaling networks within the testis [157]. A particular focus should be placed on identifying key cytokines, exosomal proteins, and metabolic mediators in blood and semen that reflect testicular immune homeostasis [158,159]. These biomarkers could serve not only for early diagnosis but also for the real-time monitoring of immune status before and after interventions, such as metabolic surgery or pharmacologic therapy. Furthermore, non-invasive diagnostic platforms based on these biomarkers would significantly improve clinical management by enabling the dynamic assessment of immune-related fertility impairments.

### 6.2. Longitudinal and Population-Based Studies of Immune Signatures

The majority of existing studies examining the link between obesity and male infertility are cross-sectional, which limits causal inference, particularly regarding immune disturbances. Future research should focus on large-scale, long-term prospective cohort studies to track how chronic obesity-induced immune dysregulation evolves over time and contributes to testicular dysfunction [160]. These studies should incorporate serial sampling of immune markers, such as inflammatory cytokines, leukocyte profiles, and oxidative stress indicators, to clarify temporal relationships. Given that most current studies predominantly involve Western populations, there is a pressing need to expand research to include diverse populations, particularly those from Asia and Africa, which would help to explore genetic or lifestyle modifiers of immune responses to obesity, thereby informing ethnically tailored therapeutic approaches.

### 6.3. Immunomodulatory Role of the Gut–Testis Axis

The gut microbiota play a pivotal role in regulating systemic and local immune homeostasis. Dysbiosis in obesity is known to amplify systemic inflammation and may contribute to immune alterations in distant organs, including the testes. Future studies should explore the gut–testis axis, focusing on how microbial metabolites—such as short-chain fatty acids—modulate testicular immune cell populations and barrier integrity [161]. Additionally, evaluating the therapeutic potential of probiotics or prebiotics in reversing dysbiosis-driven immune activation may yield novel strategies to restore immune balance and improve male reproductive outcomes.

### 6.4. Psychological and Social Determinants of Immune Dysfunction

Emerging evidence suggests that chronic psychological stress can alter immune function via neuroendocrine-immune pathways, thereby exacerbating obesity-related inflammation and potentially impairing testicular immunity [162]. Future research should examine how anxiety, depression, and social isolation influence the immune microenvironment in obese infertile men. Psychosocial interventions that reduce stress-induced immune activation may serve as adjunctive strategies to support fertility. Moreover, assessing the impact of social support and lifestyle interventions on immunological markers could help guide holistic, patient-centered treatment frameworks.

## 7. Conclusions

Obesity, along with the immune microenvironment, is closely linked to male infertility, and treatments such as interventional surgery and medical therapies, as obesity-induced changes in the immune microenvironment, significantly impact male reproductive health. The underlying mechanisms are complex, involving chronic low-grade inflammation, immune cell infiltration, and the dysregulation of adipokines such as leptin and adiponectin. These changes disrupt testicular function, impair spermatogenesis, and contribute to hormonal imbalances. A deeper understanding of how obesity influences the immune landscape within the testes and its subsequent impact on male fertility is crucial for developing more effective therapeutic strategies. Future research should prioritize investigating the specific immune pathways involved in obesity-related male infertility, with a focus on identifying novel molecular targets and exploring potential therapies to correct these immune dysregulations.

## Figures and Tables

**Figure 1 biomedicines-13-01314-f001:**
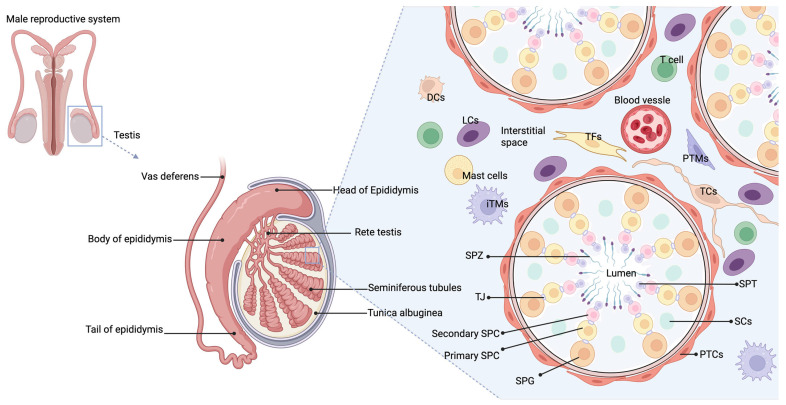
Structure of the testis.

**Figure 2 biomedicines-13-01314-f002:**
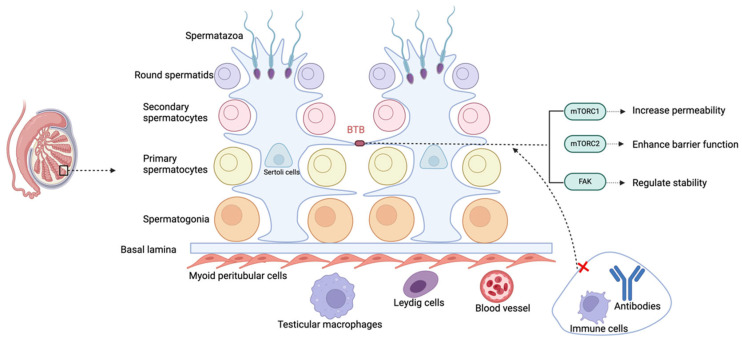
Schematic representation of the blood–testis barrier (BTB).

**Figure 3 biomedicines-13-01314-f003:**
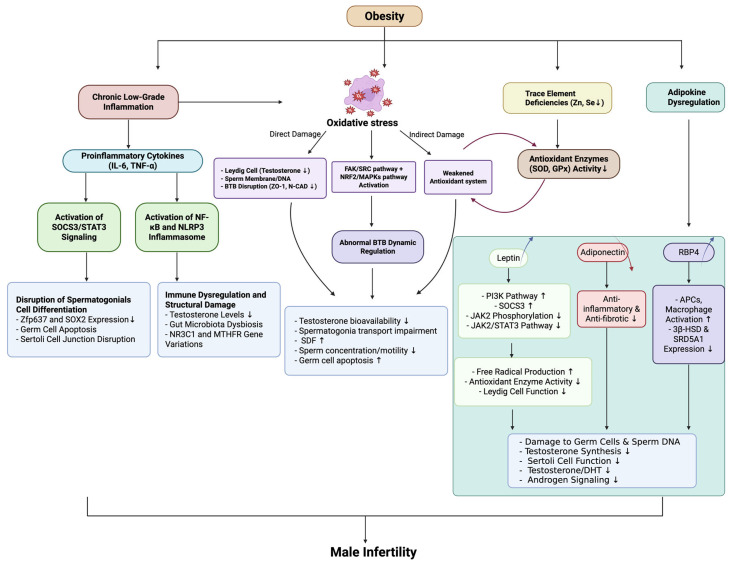
The intersection of obesity, immune microenvironment disruption, and male infertility. Abbreviations: IL-6 (Interleukin-6); TNF-α (Tumor Necrosis Factor-alpha); SOCS3 (Suppressor of Cytokine Signaling 3); STAT3 (Signal Transducer and Activator of Transcription 3); NF-κB (Nuclear Factor kappa-light-chain-enhancer of Activated B Cells); NLRP3 (NLR Family Pyrin Domain Containing 3); Zfp637 (Zinc Finger Protein 637); SOX2 (Sex-Determining Region Y-Box 2); NR3C1 (Nuclear Receptor Subfamily 3 Group C Member 1); MTHFR (Methylenetetrahydrofolate Reductase); BTB (blood–testis barrier); ZO-1 (Zona Occludens-1); N-CAD (Neural Cadherin); FAK (Focal Adhesion Kinase); SRC (Src Family Kinase); SDF (sperm DNA fragmentation); Zn (Zinc); Se (Selenium); SOD (Superoxide Dismutase); GPx (Glutathione Peroxidase); PI3K (Phosphatidylinositol 3-Kinase); JAK2 (Janus Kinase 2); APCs (Antigen-Presenting Cells); 3β-HSD (3β-Hydroxysteroid Dehydrogenase); SRD5A1 (Steroid 5α-Reductase Type 1); DHT (Dihydrotestosterone).

**Table 1 biomedicines-13-01314-t001:** Immune microenvironment dysregulation in obesity: implications for male testicular function and infertility.

Category	Mechanism	Impact on Male Reproductive Health
Chronic Inflammation	- ↑ Secretion of pro-inflammatory cytokines (e.g., IL-6, TNF-α) - ↑ Activation of inflammation signaling pathways (e.g., SOCS3/STAT3, NF-κB, NLRP3) - Gut dysbiosis → Leydig cell dysfunction - Gene polymorphisms (NR3C1, MTHFR) → Abnormal stress and inflammation-related gene expression - VC →Elevated IL-1 and TNF-α → activates NLRP3 inflammasome	- ↓ Spermatogonial cell differentiation - ↑ Germ cell apoptosis - Spermatogenesis disorders - Disrupts immune tolerance in testis
Oxidative Stress	- ↑ Overproduction of ROS - ↓ Expression of key proteins in the BTB (e.g., ZO-1, CX43) - ↓ Antioxidant system in the testes - Zn, Se, Cu, Fe deficiency → Decreased antioxidant enzyme activity	- ↓BTB integrity - ↑ SDF - ↓ Sperm motility and concentration
Hormonal Imbalance	- ↑ Aromatase expression → ↑ Conversion of androgens to estrogen	- ↓ Testosterone synthesis - ↓ Spermatogenesis
Adipokine Dysregulation	- ↑ Leptin → Activation of PI3K signaling pathway - ↓ Adiponectin → Inhibition of cAMP-PKA signaling - ↑ RBP4 → Activation of APCs and CD4^+^ T cells in adipose tissue - ↑ RBP4 → Inhibition of steroidogenic enzymes	- ↑ Sperm DNA damage and seminiferous tubule atrophy - ↑ Testicular inflammation and fibrosis - Disruption of testicular immune homeostasis - ↓ Testosterone and DHT synthesis

## Data Availability

No new data were created or analyzed in this study. Data sharing is not applicable to this article.

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
