# Peer review of "Immune Microenvironment Dysregulation: A Contributing Factor to Obesity-Associated Male Infertility"

_biomedicines, 2025, doi:10.3390/biomedicines13061314_

Round 1
Reviewer 1 Report
Comments and Suggestions for Authors
biomedicines-3577232
Immune Microenvironment Dysregulation: A Contributing Factor to Obesity-Associated Male Infertility
Rui Feng et al.,
Overview
The authors reviewed the relationship between male infertility and obesity from the viewpoint of immunologic dysregulation.
The manuscript contains clinically important topics.
The reviewer suggests some revisions and updates for improving the manuscript.
Critical comments
1. Are there any associations between genetic factors and inflammations in obese infertile men? The following paper may be some help for discussion.
Samvida S Venkatesh Genome-wide analyses identify 25 infertility loci and relationships with reproductive traits across the allele frequency spectrum. Nat Genet. 2025 Apr 14. doi: 10.1038/s41588-025-02156-8.
2. Can the authors briefly discuss the therapeutic potentials of acetate and butyrate for obese infertile men. The following papers can be some references.
Olaniyi KS, Akintayo CO, Oniyide AA, Omoaghe AO, Oyeleke MB, Fafure AA. Acetate supplementation restores testicular function by modulating Nrf2/PPAR-gamma in high fat diet-induced obesity in Wistar rats. J Diabetes Metab Disord. 2021 Oct 21;20(2):1685-1696.
Xiangen Liu, Yujuan Qi, Tao Zhu, Xiaoyue Ding, Dianshuang Zhou, Conghui Han. Butyrate improves testicular spermatogenic dysfunction induced by a high-fat diet. Transl Androl Urol. 2025 Mar 30;14(3):627-636.
3. The authors may be able to refer to the protective effects of anti-oxidative stress agents (such as resveratrol etc) on testis.
The following paper may be one of potential references.
Wang KL, Chiang YF, Huang KC, Chen HY, Ali M, Hsia SM. Alleviating 3-MCPD-induced male reproductive toxicity: Mechanistic insights and resveratrol intervention. Ecotoxicol Environ Saf. 2024 Feb;271:115978.
4. The authors also can describe the effect of icariin for improving obesity and enhancing testicular spermatogenesis.
The following paper may be a candidate reference.
Yanhong Wei, Jian Tu, Lin Ji, Rutong Wang, Runtang Zhou, Xiaocan Lei, Linlin Hu, Hua Huang. 114280Icariin inhibition of NLRP3 mediated Leydig cell pyroptosis and insulin resistance ameliorates spermatogenesis disorders in obese mice. Int Immunopharmacol. 2025 Apr 4:151:
5. Are there any reports that characterized the gut or testicular microbiome in obese infertile men?
Author Response
Comments 1: Are there any associations between genetic factors and inflammations in obese infertile men? The following paper may be some help for discussion.
Samvida S Venkatesh Genome-wide analyses identify 25 infertility loci and relationships with reproductive traits across the allele frequency spectrum. Nat Genet. 2025 Apr 14. doi: 10.1038/s41588-025-02156-8.
Response 1: Thank you for suggesting the paper by Venkatesh et al. (2025) on infertility loci and genetic factors. We have reviewed the paper, and we agree that genetic variations significantly impact inflammation pathways in the context of obesity-induced infertility. We have now included a discussion of how specific genetic loci identified in this paper may influence inflammation and contribute to the reproductive dysfunction observed in obese infertile men, which can be found on page 12 of from line 645-656, highlighted in red.
Comments 2: Can the authors briefly discuss the therapeutic potentials of acetate and butyrate for obese infertile men. The following papers can be some references.
Olaniyi KS, Akintayo CO, Oniyide AA, Omoaghe AO, Oyeleke MB, Fafure AA. Acetate supplementation restores testicular function by modulating Nrf2/PPAR-gamma in high fat diet-induced obesity in Wistar rats. J Diabetes Metab Disord. 2021 Oct 21;20(2):1685-1696.
Xiangen Liu, Yujuan Qi, Tao Zhu, Xiaoyue Ding, Dianshuang Zhou, Conghui Han. Butyrate improves testicular spermatogenic dysfunction induced by a high-fat diet. Transl Androl Urol. 2025 Mar 30;14(3):627-636.
Response 2: We greatly appreciate your suggestion to include a discussion on the therapeutic potential of acetate and butyrate. The studies by Olaniyi et al. (2021) and Liu et al. (2025) provide valuable insights into how these short-chain fatty acids influence testicular function in the context of obesity. We have expanded the manuscript to include the role of acetate and butyrate in modulating inflammation, oxidative stress, and metabolic pathways that are dysregulated in obese infertile men, which can be found on page 15 of from line 808-819, highlighted in red.
Comments 3: The authors may be able to refer to the protective effects of anti-oxidative stress agents (such as resveratrol etc) on testis. The following paper may be one of potential references.
Wang KL, Chiang YF, Huang KC, Chen HY, Ali M, Hsia SM. Alleviating 3-MCPD-induced male reproductive toxicity: Mechanistic insights and resveratrol intervention. Ecotoxicol Environ Saf. 2024 Feb;271:115978.
Response 3: We greatly appreciate your valuable suggestion. In the revised manuscript, we have fully considered your recommendation regarding the protective effects of antioxidants such as resveratrol on testicular health. We have expanded the treatment section to include an analysis of the antioxidant mechanisms of resveratrol, with a particular focus on the study by Wang et al. (2024). Additionally, we have supplemented the translational medicine section by discussing the clinical potential of these natural bioactive compounds as adjunct therapies for male infertility, which can be found on page 15 of from line 820-825, highlighted in red.
Comments 4: The authors also can describe the effect of icariin for improving obesity and enhancing testicular spermatogenesis. The following paper may be a candidate reference.
Yanhong Wei, Jian Tu, Lin Ji, Rutong Wang, Runtang Zhou, Xiaocan Lei, Linlin Hu, Hua Huang. 114280Icariin inhibition of NLRP3 mediated Leydig cell pyroptosis and insulin resistance ameliorates spermatogenesis disorders in obese mice. Int Immunopharmacol. 2025 Apr 4:151:
Response 4: We thank you for highlighting the study by Wei et al. (2025) on the role of icariin in improving obesity and enhancing spermatogenesis. We have now included a discussion of icariin’s mechanism of action, offering valuable insights into its potential as a therapeutic agent for improving testicular function and spermatogenesis in obese infertile men. This addition can be found on page 15, lines 820–825, highlighted in red.
Comments 5: Are there any reports that characterized the gut or testicular microbiome in obese infertile men?
Response 5: We agree that exploring the microbiome—both gut and testicular—may offer novel insights into the pathophysiology of obesity-related male infertility. While the gut microbiome has been extensively studied in relation to metabolic disorders, its direct impact on male reproductive health remains insufficiently understood. In response to your suggestion, we have added a brief discussion on the current status of microbiome research, referencing studies that investigate the potential connection between gut dysbiosis and testicular dysfunction in obese men. Although direct evidence on the testicular microbiome in obese infertile men is currently limited, we have emphasized this emerging area as a promising direction for future research. This addition can be found on page 12, lines 631–644.
Reviewer 2 Report
Comments and Suggestions for Authors
1. The authors have prepared a literature review describing how obesity-induced alterations in the immune microenvironment can result in male infertility. However, numerous high-quality review articles—whether narrative reviews, systematic reviews, or meta-analyses—have already been published on this topic. The authors must clarify the novelty of their current review in comparison to those existing works and specify how it distinguishes itself from prior publications. Unless the manuscript clearly demonstrates such novelty within the main text, it may be difficult to justify publication in this journal.
To this end, the authors are encouraged to thoroughly examine and, where appropriate, cite the following relevant review articles. Please make explicit in the text how your review differs from these prior works:
-
Cannarella R, et al. Mol Aspects Med. 2024. PMID: 38593513 (Narrative Review)
-
Santi D, et al. Andrology. 2024. PMID: 37226894 (Meta-Analysis)
-
Service CA, et al. Fertil Steril. 2023. PMID: 37839720 (Systematic Review)
-
George BT, et al. Int J Mol Sci. 2023. PMID: 38203349 (Narrative Review)
-
Peel A, et al. Andrology. 2023. PMID: 36789664 (Narrative Review)
-
Salas-Huetos A, et al. Obes Rev. PMID: 32705766 (Systematic Review and Meta-Analysis)
-
Kahn BE, et al. Curr Opin Urol. 2017. PMID: 28661897 (Narrative Review)
-
Liu Y, et al. Reproduction. 2017. PMID: 28747541 (Narrative Review)
-
Campbell JM, Reprod Biomed Online. 2015. PMID: 26380863 (Systematic Review and Meta-Analysis)
2. The manuscript’s structure is quite clear: the second section introduces general information on the immune microenvironment and male reproductive systems, the third section discusses the detrimental effects of obesity on the immune system, and the fourth section integrates findings to explain how obesity exerts its negative impact on male fertility via the immune microenvironment. Because this progression is easy to follow, it would be helpful for readers if, in the Introduction, you provide an overview of these organizational steps. A concise roadmap will give readers a smoother understanding of the review’s logical flow.
3. Consider providing a graphical abstract to give an at-a-glance overview of the paper’s central theme. In particular, since the third and fourth sections emphasize the importance of visual understanding, an impactful figure that integrates how obesity leads to male infertility through alterations in the immune microenvironment would be ideal. Currently, there appear to be few figures in these sections.
4. Varicocele is discussed only in the Introduction but does not appear in the rest of the manuscript. This may confuse readers about the main topic. Since the latter part of the Introduction indicates that the central themes of this study are chronic inflammation, cytokine imbalance, and increased reactive oxygen species, please consider either omitting varicocele altogether or, if it is relevant, referring to it separately in a more suitable section.
5. During the review, I became interested in the authors’ motivation for writing this article and attempted to look up more details about the “Medical and Health Science and Technology Project of Zhejiang Province (No. C-2024-w1178)” but was unable to find specific information. While the authors are certainly correct to avoid excessive self-citation for ethical reasons, it might be helpful to include, however briefly, an explanation in the Introduction regarding the authors’ own background (e.g., what prior work underpins this review) and how this paper fits into the larger context of the cited project. Even a short mention of the motivation behind the review could clarify its purpose.
6. Figure 1, showing a schematic of the microscopic structure of the testis, appears to be substantially inaccurate to an extent that may be educationally misleading. Specific issues include: it is unclear which elements are meant to represent Sertoli cells; seminiferous tubules are depicted as a homogeneous population (whereas, in reality, they display distinct heterogeneity depending on the cycle of the seminiferous epithelium—traditionally recognized as having six stages, PMID: 25905260); the interstitial tissue appears disproportionately large; and it remains unclear which anatomical structure is intended to be represented as being covered (possibly the tunica albuginea?).
Even an illustration by a college student, such as the one available at
https://www.linkedin.com/posts/premieka-muthukumar-2638b4221_histologyoftestis-anatomy-histology-activity-7182561223278800896-IZW8
might be more accurate in representing testicular histology. For the purposes of your article, please consider revising or replacing Figure 1 so it does not cause educational misunderstandings.
7. Please ensure that all abbreviations used in each figure are clearly defined within the corresponding figure legend so that the reader does not have to reference previous figures to find definitions. For instance, Figure 3’s legend explains BTB, but Figure 2’s legend does not. Abbreviations such as BTB, TLR, MIF, etc., should be defined each time they appear in a new figure.
8. It may be beneficial to briefly mention the role of trace elements such as zinc and copper, which are known to protect sperm from oxidative stress. Inspiration can be found in studies investigating metabolic surgery for obesity (DOI: 10.3390/nu12113354) and in the broader literature on the involvement of trace elements in immune system homeostasis. Indeed, a metallomics-based perspective is gaining attention both in obesity (DOI: 10.1002/biof.1946; DOI: 10.3390/nu15102347) and in male infertility (DOI: 10.22514/j.androl.2025.013).
For instance, zinc, nickel, and cobalt have been reported to suppress the expression of inflammatory genes in adipocytes (DOI: 10.1039/c3mt20209g), while the adipokine ZAG is known to promote lipid mobilization and reduce inflammation (DOI: 10.1007/s12011-019-01702-w). These points may be relevant to explaining the effect of obesity on male infertility through the immune microenvironment. Similarly, high doses of molybdenum may induce lipid peroxidation affecting sperm quality (DOI: 10.3109/19396368.2013.791347), and excessive phosphorus may harm reproductive function via oxidative stress (DOI: 10.1002/rmb2.12584). Although these are not essential to include, you may wish to incorporate them if you believe they will deepen the discussion.
9. The fifth section of your manuscript deals with therapeutic strategies in clinical practice. These include lifestyle interventions aimed at resolving obesity and improving semen parameters, as well as antioxidant supplementation for chronic inflammatory diseases. Historically, the presence of leukocytes in the semen has been used to evaluate chronic inflammation, and more recently, measurement of oxidative-reduction potential (ORP) in seminal plasma has been investigated. It would be valuable to discuss whether reducing obesity might improve these parameters. If there are any relevant studies, citing them could enhance your discussion.
For example, there are findings that correlate increased leukocytes in semen with lipid peroxidation of the sperm membrane (DOI: 10.1016/j.fertnstert.2006.12.025), that obese men have high levels of ROS in sperm and lower mitochondrial membrane potential (DOI: 10.1002/2211-5463.13589), and that a combined lifestyle intervention with antioxidants plus exercise did not improve ORP, but did reduce DFI (DOI: 10.1590/S1677-5538.IBJU.2021.0604).
10. The sixth section feels somewhat miscellaneous and abrupt. Since the main topic of the article is the immune microenvironment in obese patients, it would be helpful to explain in greater detail how “population-based studies,” the “gut-testis axis,” and “psychological and social factors” pertain to this immune-related pathology. More explicit connections would likely make your discussion more coherent.
Author Response
Comments 1: The authors have prepared a literature review describing how obesity-induced alterations in the immune microenvironment can result in male infertility. However, numerous high-quality review articles—whether narrative reviews, systematic reviews, or meta-analyses—have already been published on this topic. The authors must clarify the novelty of their current review in comparison to those existing works and specify how it distinguishes itself from prior publications. Unless the manuscript clearly demonstrates such novelty within the main text, it may be difficult to justify publication in this journal.
To this end, the authors are encouraged to thoroughly examine and, where appropriate, cite the following relevant review articles. Please make explicit in the text how your review differs from these prior works:
- Cannarella R, et al. Mol Aspects Med. PMID: 38593513(Narrative Review)
- Santi D, et al. 2024. PMID: 37226894 (Meta-Analysis)
- Service CA, et al. Fertil Steril. PMID: 37839720(Systematic Review)
- George BT, et al. Int J Mol Sci. PMID: 38203349(Narrative Review)
- Peel A, et al. 2023. PMID: 36789664 (Narrative Review)
- Salas-Huetos A, et al. Obes Rev.PMID: 32705766 (Systematic Review and Meta-Analysis)
- Kahn BE, et al. Curr Opin Urol. PMID: 28661897(Narrative Review)
- Liu Y, et al. 2017. PMID: 28747541 (Narrative Review)
- Campbell JM, Reprod Biomed Online. PMID: 26380863(Systematic Review and Meta-Analysis)
Response 1: Thank you for the valuable comments. We highly appreciate the important reviews that already exist in this field and have carefully studied the literature suggested by the reviewer. To highlight the uniqueness of our review, we have added a paragraph in the Introduction (see Page2–3, Lines 126–159, marked in red) to clearly outline the novelty of our work.
Comments 2: The manuscript’s structure is quite clear: the second section introduces general information on the immune microenvironment and male reproductive systems, the third section discusses the detrimental effects of obesity on the immune system, and the fourth section integrates findings to explain how obesity exerts its negative impact on male fertility via the immune microenvironment. Because this progression is easy to follow, it would be helpful for readers if, in the Introduction, you provide an overview of these organizational steps. A concise roadmap will give readers a smoother understanding of the review’s logical flow.
Response 2: We appreciate the reviewer’s suggestion. We have added a concise paragraph at the end of the Introduction (Page 3, Lines 160-166, marked in red) that outlines the structure of the review, highlighting the progression from the immune system and reproductive physiology to obesity-induced immune alterations, and finally to their impact on male fertility. This roadmap aims to improve the reader’s understanding of the logical flow of the manuscript.
Comments 3: Consider providing a graphical abstract to give an at-a-glance overview of the paper’s central theme. In particular, since the third and fourth sections emphasize the importance of visual understanding, an impactful figure that integrates how obesity leads to male infertility through alterations in the immune microenvironment would be ideal. Currently, there appear to be few figures in these sections.
Response 3: Thank you for this constructive suggestion. We agree that a graphical abstract would greatly enhance the clarity of our review. We have designed and added a new graphical abstract (now Figure 3) that illustrates how obesity affects male fertility through alterations in the immune microenvironment, including mechanisms such as inflammation activation, changes in immune cell subsets, and increased oxidative stress. We believe this integrative figure will enhance readers’ overall understanding of the key mechanisms.
Comments 4: Varicocele is discussed only in the Introduction but does not appear in the rest of the manuscript. This may confuse readers about the main topic. Since the latter part of the Introduction indicates that the central themes of this study are chronic inflammation, cytokine imbalance, and increased reactive oxygen species, please consider either omitting varicocele altogether or, if it is relevant, referring to it separately in a more suitable section.
Response 4: We agree that varicocele should be more appropriately placed in a relevant section rather. Since it shares mechanistic links with chronic inflammation in obesity man, we have relocated the discussion of varicocele to a more suitable part of the manuscript (see Pages 11–12, Lines 621–630, highlight in red), where we elaborate on its potential interaction with immune dysregulation.
Comments 5: During the review, I became interested in the authors’ motivation for writing this article and attempted to look up more details about the “Medical and Health Science and Technology Project of Zhejiang Province (No. C-2024-w1178)” but was unable to find specific information. While the authors are certainly correct to avoid excessive self-citation for ethical reasons, it might be helpful to include, however briefly, an explanation in the Introduction regarding the authors’ own background (e.g., what prior work underpins this review) and how this paper fits into the larger context of the cited project. Even a short mention of the motivation behind the review could clarify its purpose.
Response 5: Thank you for the reviewer’s attention to the background of our study. We have added a brief explanation in the introduction. Please refer to Page 3, Lines 139–141, where the modifications are marked in red.
Comments 6: Figure 1, showing a schematic of the microscopic structure of the testis, appears to be substantially inaccurate to an extent that may be educationally misleading. Specific issues include: it is unclear which elements are meant to represent Sertoli cells; seminiferous tubules are depicted as a homogeneous population (whereas, in reality, they display distinct heterogeneity depending on the cycle of the seminiferous epithelium—traditionally recognized as having six stages, PMID: 25905260); the interstitial tissue appears disproportionately large; and it remains unclear which anatomical structure is intended to be represented as being covered (possibly the tunica albuginea?).
Even an illustration by a college student, such as the one available at
https://www.linkedin.com/posts/premieka-muthukumar-2638b4221_histologyoftestis-anatomy-histology-activity-7182561223278800896-IZW8
might be more accurate in representing testicular histology. For the purposes of your article, please consider revising or replacing Figure 1 so it does not cause educational misunderstandings.
Response 6: Thank you for pointing out the inaccuracies in Figure 1. We have completely revised this figure to more accurately represent the microscopic anatomy of the testis.
Comments 7: Please ensure that all abbreviations used in each figure are clearly defined within the corresponding figure legend so that the reader does not have to reference previous figures to find definitions. For instance, Figure 3’s legend explains BTB, but Figure 2’s legend does not. Abbreviations such as BTB, TLR, MIF, etc., should be defined each time they appear in a new figure.
Response 7: We appreciate this important detail. We have now revised all figure legends to ensure that all abbreviations are clearly defined, and we have also reviewed and clarified all abbreviations used throughout the manuscript.
Comments 8: It may be beneficial to briefly mention the role of trace elements such as zinc and copper, which are known to protect sperm from oxidative stress. Inspiration can be found in studies investigating metabolic surgery for obesity (DOI: 10.3390/nu12113354) and in the broader literature on the involvement of trace elements in immune system homeostasis. Indeed, a metallomics-based perspective is gaining attention both in obesity (DOI: 10.1002/biof.1946; DOI: 10.3390/nu15102347) and in male infertility (DOI: 10.22514/j.androl.2025.013).
For instance, zinc, nickel, and cobalt have been reported to suppress the expression of inflammatory genes in adipocytes (DOI: 10.1039/c3mt20209g), while the adipokine ZAG is known to promote lipid mobilization and reduce inflammation (DOI: 10.1007/s12011-019-01702-w). These points may be relevant to explaining the effect of obesity on male infertility through the immune microenvironment. Similarly, high doses of molybdenum may induce lipid peroxidation affecting sperm quality (DOI: 10.3109/19396368.2013.791347), and excessive phosphorus may harm reproductive function via oxidative stress (DOI: 10.1002/rmb2.12584). Although these are not essential to include, you may wish to incorporate them if you believe they will deepen the discussion.
Response 8: We thank the reviewer for this excellent suggestion. In the section 4, we have now included a paragraph summarizing the potential role of trace elements such as zinc, copper, and cobalt in modulating oxidative stress and immune responses. We discuss how imbalances in these elements may contribute to the infertility phenotype seen in obese males. The modifications can be found on Page 13, Lines 685–703.
Comments 9: The fifth section of your manuscript deals with therapeutic strategies in clinical practice. These include lifestyle interventions aimed at resolving obesity and improving semen parameters, as well as antioxidant supplementation for chronic inflammatory diseases. Historically, the presence of leukocytes in the semen has been used to evaluate chronic inflammation, and more recently, measurement of oxidative-reduction potential (ORP) in seminal plasma has been investigated. It would be valuable to discuss whether reducing obesity might improve these parameters. If there are any relevant studies, citing them could enhance your discussion. For example, there are findings that correlate increased leukocytes in semen with lipid peroxidation of the sperm membrane (DOI: 10.1016/j.fertnstert.2006.12.025), that obese men have high levels of ROS in sperm and lower mitochondrial membrane potential (DOI: 10.1002/2211-5463.13589), and that a combined lifestyle intervention with antioxidants plus exercise did not improve ORP, but did reduce DFI (DOI: 10.1590/S1677-5538.IBJU.2021.0604).
Response 9: We appreciate this insightful comment. The section has been revised as suggested. Please refer to page 15, lines 801-807, where the changes have been incorporated.
Comments 10: The sixth section feels somewhat miscellaneous and abrupt. Since the main topic of the article is the immune microenvironment in obese patients, it would be helpful to explain in greater detail how “population-based studies,” the “gut-testis axis,” and “psychological and social factors” pertain to this immune-related pathology. More explicit connections would likely make your discussion more coherent.
Response 10: Thank you for this helpful comment. We have restructured this section to clearly articulate how "population-based epidemiological studies," the "gut–testis axis," and "psychosocial factors" are linked to obesity-induced immune dysregulation. By refining the logic and section headings, we aim to enhance the coherence and overall clarity of this part. This change can be found on page 16-17, lines 952-1092, highlight in red.
Round 2
Reviewer 1 Report
Comments and Suggestions for Authors
The manuscript markedly improved.
Author Response
Comment 1: The manuscript markedly improved.
Response 1: Thank you for your kind words. We appreciate your feedback and are pleased that the revisions improved the manuscript.
Reviewer 2 Report
Comments and Suggestions for Authors
I believe that the manuscript’s overall quality has improved significantly thanks to the extensive revisions, and I would like to commend the authors for their efforts. After carefully reviewing the revised manuscript, I noted the following point for your attention:
You added the statement, “In parallel, dysfunction of metal ion transporters such as the Zinc Transporter (ZIP) family disrupts the homeostasis of trace element distribution, leading to imbalances in seminal plasma components, especially Zn and Se secreted by the prostate [126].” However, it appears that Reference [126] does not discuss any imbalance of zinc and selenium secreted by the prostate. Could you please double-check whether this is the appropriate reference, or consider providing a more suitable one if necessary?
Author Response
Comments 1: I believe that the manuscript’s overall quality has improved significantly thanks to the extensive revisions, and I would like to commend the authors for their efforts. After carefully reviewing the revised manuscript, I noted the following point for your attention:
You added the statement, “In parallel, dysfunction of metal ion transporters such as the Zinc Transporter (ZIP) family disrupts the homeostasis of trace element distribution, leading to imbalances in seminal plasma components, especially Zn and Se secreted by the prostate [126].” However, it appears that Reference [126] does not discuss any imbalance of zinc and selenium secreted by the prostate. Could you please double-check whether this is the appropriate reference, or consider providing a more suitable one if necessary?
Response 1: Thank you for your careful review and valuable comment. We sincerely appreciate your attention to detail regarding the reference accuracy. Upon rechecking, we acknowledge that Reference [126] (originally cited) does not explicitly discuss the imbalance of zinc (Zn) and selenium (Se) in seminal plasma due to ZIP transporter dysfunction. To address this, we have revised the sentence to focus on the role of zinc transporters (ZIPs) in systemic and testicular zinc homeostasis, supported by a more appropriate reference (now cited as [124] in the revised manuscript). Please refer to Page 16, Lines 691-694. We hope this revision aligns better with the cited literature and clarifies the mechanistic link between zinc transporter dysfunction and male reproductive impairment. Thank you for highlighting this issue.
We apologize for the oversight and have ensured that all citations now accurately reflect the discussed mechanisms.